# Elementary steps in electrical doping of organic semiconductors

Max L. Tietze[1,2,4], Johannes Benduhn[1], Paul Pahner[1], Bernhard Nell[1], Martin Schwarze [1], Hans Kleemann[1], Markus Krammer[3], Karin Zojer[3], Koen Vandewal[1,5] & Karl Leo[1]

Fermi level control by doping is established since decades in inorganic semiconductors and has been successfully introduced in organic semiconductors. Despite its commercial success in the multi-billion OLED display business, molecular doping is little understood, with its elementary steps controversially discussed and mostly-empirical-materials design. Particularly puzzling is the efficient carrier release, despite a presumably large Coulomb barrier. Here we quantitatively investigate doping as a two-step process, involving single-electron transfer from donor to acceptor molecules and subsequent dissociation of the ground-state integer-charge transfer complex (ICTC). We show that carrier release by ICTC dissociation has an activation energy of only a few tens of meV, despite a Coulomb binding of several 100 meV. We resolve this discrepancy by taking energetic disorder into account. The overall doping process is explained by an extended semiconductor model in which occupation of ICTCs causes the classically known reserve regime at device-relevant doping concentrations.

[1] Dresden Integrated Center for Applied Physics and Photonic Materials, Technische Universität Dresden, Nöthnitzer Strasse 61, 01187 Dresden, Germany. [2] Physical Science and Engineering Division, KAUST Solar Center, King Abdullah University of Science and Technology, Thuwal 23955-6900, Saudi Arabia. [3] NAWI Graz, Institute of Solid State Physics, Graz University of Technology, Petersgasse 16, 8010 Graz, Austria. [4] Present address: Department of Microbial and Molecular Systems, Centre for Surface Chemistry and Catalysis, KU Leuven—University of Leuven, Celestijnenlaan 200F, B-3001 Leuven, Belgium. [5] Present address: Instituut voor Materiaalonderzoek, Hasselt University, Wetenschapspark 1, 3590 Diepenbeek, Belgium. Correspondence and requests for materials should be addressed to M.L.T. (email: max.tietze@iapp.de) or to K.L. (email: karl.leo@iapp.de)

Molecular doping is a powerful technique to precisely control the electronic properties of organic devices such as light-emitting diodes (OLEDs)[1], solar cells[2], field effect transistors[3,4], thermo-electric generators[5], and photo-detectors[6]. Furthermore, redox-active organic compounds were recently suggested for sustainable battery systems[7,8]. Although commercially employed in OLED displays, the working principle of molecular doping is still a controversial topic[9–12], hindering the development of novel redox couples[13,14]. As initial step, either host-dopant electronic wave-function hybridization or ground-state integer-charge transfer (ICT) from donor (D) to acceptor (A) molecules were identified as fundamental mechanisms[15,16]. Subsequent steps are only qualitatively understood. Experimentally, research mostly focused on certain aspects such as trap-filling[9,10,17–19] or density-of-states (DOS) modification[20] upon doping. Comprehensive hopping transport simulations have aimed at describing the orders of magnitude conductivity enhancements[21–23]. In particular, the super-linear scaling with the dopant concentration is puzzling[12,21–25]. In that context, the crucial intermediate step of carrier release after ICT is not sufficiently understood, although investigated since two decades[26].

The essence of an electronic doping effect is a shift of the Fermi level $E_F$ toward the highest occupied (p-doping) or lowest unoccupied (n-doping) states[27,28], with the position of $E_F$ linked to the overall doping efficiency, $\eta_{dop} = p/N_A$ (p-doping). The free carrier $p$, neutral $N_A$, and ionized $N_A^-$ dopant densities are linked by the charge neutrality condition known from classical semiconductor physics[29]

$$p = N_A^- = \frac{N_A}{1 + \exp\left(\frac{E_A - E_F}{k_B T}\right)} \qquad (1)$$

In recent times, we have shown that this approach explains the Fermi level shift and doping efficiency of various molecularly doped organic semiconductors, measured by ultraviolet photo-electron spectroscopy (UPS) at room temperature (RT)[9,10]. Most importantly, it was argued that the organic systems are forced into the classical reserve regime due to Fermi level pinning at the acceptor level $E_A$, explaining commonly observed low doping efficiencies of < 10%.[9,30–34].

However, the thermal activation behavior and physical origin of the acceptor level $E_A$ in organic materials have not clearly been addressed yet, although being required for a complete picture. Furthermore, validity of Eq. (1) is not necessarily given for organic materials, as Coulomb binding energies are much higher (typically 0.5 eV $\gg k_B T$ at RT) than in inorganic single-crystal semiconductors. Consequently, formed ground-state ICT complexes (ICTCs) must be separated for yielding free charge carriers $p$ (see Fig. 1a)[9,12], referring to polarons in organic materials[35]. Besides thermal dissociation, ICTC separation is controlled by the (local) energetic landscape and related electrostatic interactions[12,36–38].

Here we directly compare the thermal activation of ICT and free carriers in archetypical p-doped organic semiconductors by means of deliberately chosen complementary experimental approaches in a wide temperature range (10 K < T < 300 K). The respective activation energies are found to be different, calling for a generalization of Eq. (1) in organic materials. $C(V)$ spectroscopy on Schottky diodes indicates a gradual freeze-out of the free carrier density $p(T)$ until complete device depletion at 20 K. In contrast, absorption measurements reveal that host-dopant ICT remains temperature independent, even at 10 K. We conclude that dissociation of $[D^+A^-]$ ICTCs is the predominant process controlling the thermal release of mobile polarons. These findings can be consistently understood in terms of an extended statistical description, including the Coulomb binding of carriers within ICTCs, $E_{CT}^b$, in addition to the classical (shallow) acceptor level $E_A$. Although $E_{CT}^b$ comprises several 100 meV and is assigned to cause Fermi level pinning, the thermal Arrhenius-type activation for hole release determined by Mott–Schottky analysis is only a few 10 meV and can be even such low as 9 meV as obtained for the prototypical OLED hole transporter system N,N,N',N'-Tetrakis(4-methoxyphenyl)-benzidine (MeO-TPD):1,3,4,5,7,8-hexafluorotetracyanonaphthoquinodimethane (F$_6$-TCNNQ). This effective lowering is ascribed to originate from energetic disorder. The presence of the freeze-out regime is confirmed by resolving depletion widths at metal/organic contacts by incremental UPS for wide ranges of doping concentrations and temperatures. Finally, the super-linear conductivity scaling is consistently modeled by Monte Carlo transport simulations, taking Coulomb interactions between ionized dopants and charge carriers into account.

Our findings yield a consistent picture of molecular doping, showing particularly that the freeze-out (reserve) regime of the doping efficiency is directly linked to the thermally activated release of charge carriers from ICTCs after single-electron transfer. The development of efficient host-dopant systems should therefore not solely aim for synthesis of strong dopant compounds in terms of energy levels but also for optimized interface energetics regarding electrostatics and disorder.

## Results

**Temperature-independent polaron absorption.** Ground-state ICT is characterized by polaron absorption, which yields typical

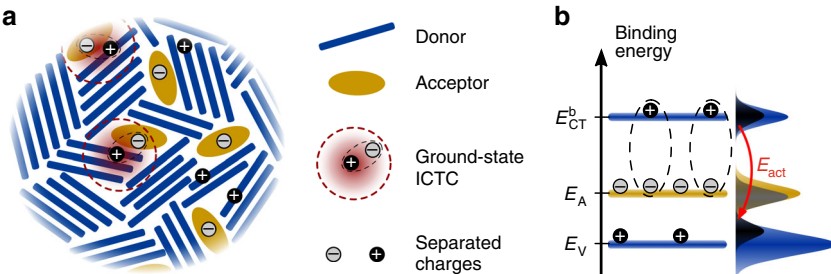

**Fig. 1** Sketch of the two-step process of molecular doping. **a** Illustration of charged species in a p-doped organic semiconductor film: integer-charge transfer from donor to acceptor molecules, DA → $[D^+A^-]$, yields ground-state integer-charge transfer complexes (ICTCs), which are thermally dissociated into separated charges. **b** Sketch of a statistical two- or complete three-level model for p-doping. Black and gray distributions indicate the occupations of Gaussian broadened levels $E_V$ or $E_{CT}^b$ (each blue) with holes and acceptor states $E_A$ (yellow) with electrons, respectively. The energy required for ICTC dissociation is reduced from $E_{CT}^b$—$E_V$ to $E_{act}$ due to energetic disorder, i.e., from several 100 mV to effectively only a few tens of meV. ICT itself is temperature-independent. Incomplete ICT occurs if $IE(D) > EA(A)$, which is expressed by $E_A$ describing the degree of acceptor ionization in equilibrium

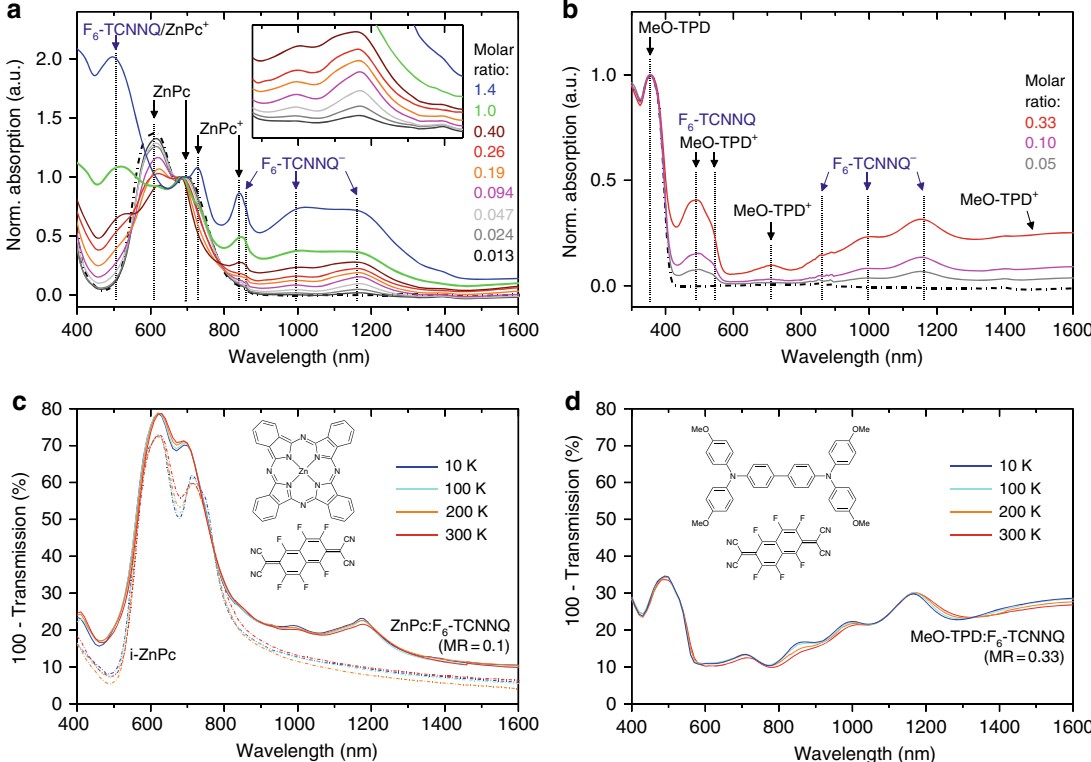

**Fig. 2** Absorption spectroscopy on p-doped films. Normalized absorption spectra of **a** ZnPc:$F_6$-TCNNQ and **b** MeO-TPD:$F_6$-TCNNQ thin films of varying doping ratio measured at RT, evidencing integer-charge transfer (ICT) each. **c**, **d** Respective transmission spectra of representative doped films under temperature variation ($T = 10...\,300\,K$). Spectra of undoped ZnPc and MeO-TPD films are given as dash-dotted lines. Energy values are as follows: $IE_{UPS}(ZnPc) = 5.0\,eV$, $IE_{UPS}(MeO\text{-}TPD) = 5.1\,eV$, and $EA_{DFT}(F_6\text{-}TCNNQ) = 5.2\,eV$[15], the corresponding molecular structures are given in panel **c** and **d**

(sub-gap) transitions, i.e., $D^+ \rightarrow D^{+\star}$ and $A^- \rightarrow A^{-\star}$ excitations[39–41]. This fingerprint is distinct from possibly formed supramolecular $D^{+\delta}A^{-\delta}$ complexes with fractional charge transfer (CT), $\delta < 1$, whose absorption is characterized by optical transitions of the form $D^{+\delta}A^{-\delta} \rightarrow D^{+1-\delta}A^{-1+\delta}$ [39,42].

Normalized absorption spectra of the small molecule systems ZnPc:$F_6$-TCNNQ and MeO-TPD:$F_6$-TCNNQ, examined at RT, are shown in Fig. 2a, b. Undoped reference samples (dash-dotted lines) do not show sub-gap features. For ZnPc, only the characteristic Q-band absorption peaks at 610 and 700 nm are present. For the highly transparent MeO-TPD, the absorption is negligibly weak for wavelengths > 410 nm. Upon p-doping, sub-gap absorption peaks at ∼ 860, 995, and 1160 nm appear for both hosts, with intensities scaling with doping ratio, attributed to $F_6$-TCNNQ$^-$ anions[43]. The energetic distance of the ZnPc Q-bands reduces, accompanied by a decrease of the π–π interaction peak (610 nm)[44], which indicates suppressed crystal-phase formation[21]. For the most heavily doped ZnPc films, two additional features (at ∼730 and 840 nm) are attributed to integer-charged ZnPc$^+$ molecules (see Nyokong et al.[45]). Apart from that, absorption of neutral $F_6$-TCNNQ molecules is found (520 nm peak, Supplementary Figure 1), indicating an incomplete host-dopant ICT at higher doping ratios. For p-doped MeO-TPD films, various sub-gap absorption features appear besides the $F_6$-TCNNQ$^-$ features, with distinct peaks at 490, 545, and 710 nm attributed to MeO-TPD$^+$ cations[41]. In contrast to p-ZnPc, absorption at wavelengths longer than 1300 nm remains high, indicating enhanced MeO-TPD$^+$ polaron delocalization[46].

We now study the relative degree of ICT under temperature variation, shown in Fig. 2c, d for representative p-doped films of each host. The temperature is step-wise reduced to 10 K. Importantly, no remarkable intensity drop is noticeable for the

$F_6$-TCNNQ$^-$ features in both hosts. On the contrary, the $F_6$-TCNNQ$^-$ absorptance even seems to slightly increase. Similar findings hold for the cationic host absorptions, clearly visible for MeO-TPD$^+$ (cf. 490 and 710 nm features), and for lower and higher doping ratios (Supplementary Figure 1). Therefore, we conclude that the degree of dopant ionization, i.e, host-dopant ICT, is not (or negligibly weak) temperature activated, in particular for D:A systems with $EA(A) > IE(D)$ as studied here.

**Arrhenius-type free charge-carrier activation.** To estimate doping efficiency and free hole activation $p(T)$, Mott–Schottky analysis[18] under temperature variation is performed on indium tin oxide (ITO)/host:$F_6$-TCNNQ(50 nm)/aluminum(Al) diodes utilizing ZnPc and MeO-TPD. The depletion capacitance $C_d$ of the metal/organic contact is given by:

$$\frac{d}{dV}\frac{1}{C_d^2} = \frac{2}{e\varepsilon_0\varepsilon_r A^2}\frac{1}{N_{A,d}^-} \quad . \tag{2}$$

Here, $N_{A,d}^-$ denotes the density of ionized acceptors contributing to formation of the space charge layer due to depletion of free holes $p$. The $1/C_d^2(V)$ plots for $150 < T < 290\,K$ are given in Supplementary Figure 3. Applying Eq. (2) for each material system yields decreasing doping efficiencies with decreasing temperature. More precisely, linear $\ln(N_{A,d}^-)$ vs. $T^{-1}$ Arrhenius activation of the free carrier density

$$p = N_{A,d}^- \propto \exp\left(-\frac{E_{act}}{k_B T}\right) \tag{3}$$

is found with $E_{act} = 20.7\,meV$ for p-ZnPc and 9.1 meV for p-MeO-TPD (cf. Figure 3a, b). These values are in the same order as

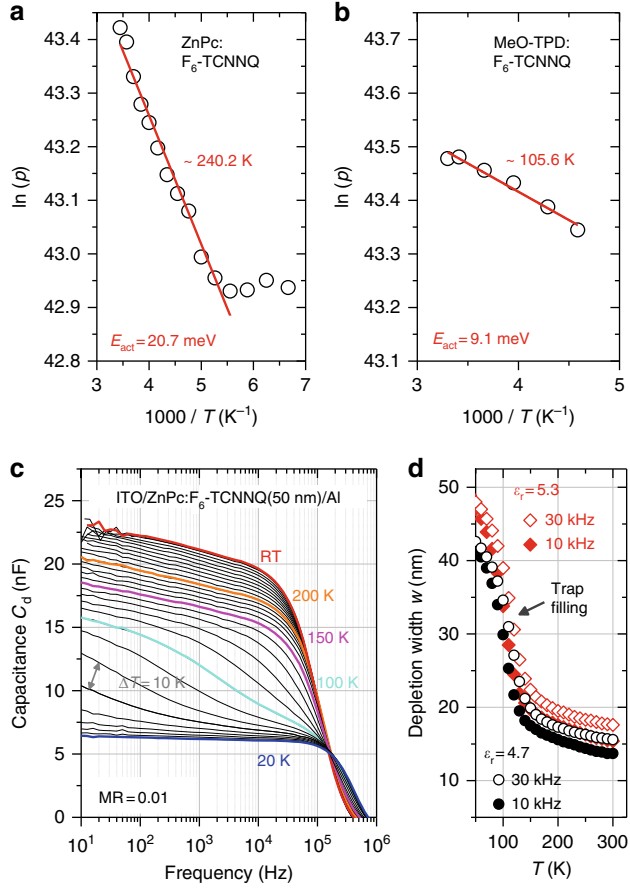

**Fig. 3** Impedance spectroscopy on p-doped films. Arrhenius-type doping activation determined by Mott-Schottky analysis on ITO/host:F$_6$-TCNNQ (50 nm)/Al diodes with **a** ZnPc or **b** MeO-TPD as host, respectively. **c** $C(f)$ spectra of the ZnPc:F$_6$-TCNNQ device at zero bias and varying sample temperature from RT to 20 K in steps of $\Delta T = 10$ K. Doping freeze-out causes steadily increasing depletion and, for $T \leq 100$ K, vanishing trap response. **d** Depletion width $w$ calculated from (**b**) at 10 and 30 kHz, and $\varepsilon_r$ = {4.7; 5.3}

**Table 1 Compilation of Arrhenius-type activation energies**

|  | F$_6$-TCNNQ | C$_{60}$F$_{36}$ |
| --- | --- | --- |
| MeO-TPD | 9.1 | 16[9] |
| Ir(piq)$_3$ | 9.5 | |
| Pentacene (P5) | 19[18] | 54[18] |
| ZnPc | 21 | |

Summary of activation energy values in meV determined by $T$-dependent Mott–Schottky analysis on ITO/host:dopant (~ 0.5 wt%)/Al structures

previously determined (cf. Table 1)[9,18,47]. Unexpectedly, the amorphous MeO-TPD doped by F$_6$-TCNNQ yields the by far lowest activation energy and hence the highest doping efficiency at RT. The absolute ratio $N_{A,d}^-/N_A$ decreases from 0.62 at 290 K to 0.54 at 218 K, whereas for ZnPc:F$_6$-TCNNQ it drops more from 0.44 at 290 K to 0.28 at 190 K (cf. Supplementary Figure 3).

At lower temperatures, Mott–Schottky analysis becomes unreliable due to an increased influence of traps, illustrated in Fig. 3c, showing $C_d(f,T)$ spectra for ZnPc:F$_6$-TCNNQ. Below 150 K, the depletion capacitance dramatically drops, as the density of free carriers provided by doping is not sufficient to fill the (deep) traps anymore[10]. For $T < 100$ K, the trap response itself vanishes ($f < 100$ Hz)[18] and at 20 K the $C_d(f)$ curve is flat. The Schottky

diode is completely depleted. This gradual freeze-out is further illustrated by estimating the respective depletion width, $w = \varepsilon_0\varepsilon_r A/C_d$, which drastically increases below 150 K (cf. Figure 3d).

**Introducing ICTCs into the dopant-host occupation statistics**. Comparing the above discussed temperature dependence of the polaron absorption with the free carrier density strongly suggests that the doping freeze-out is solely due to a reduction in the density of free carriers $p(T)$ rather than ionized acceptor molecules. Therefore, we conclude that the observed Arrhenius-type activation energies $E_{act} \approx \{9; 21\}$ meV correspond to the energies required for dissociating [D$^+$A$^-$] ICTCs into mobile polarons and acceptor anions A$^-$. This conclusion is very surprising, as $E_{act}$ is, first, much smaller than the expected Coulomb binding energy of an ICTC ($E_{CT}^b \approx 0.5$ eV), and second, much smaller than typical Arrhenius-type activation energies of conductivities at medium doping ratios such as the 1 mol% used here ($E_{act,\sigma} \approx 100...330$ meV)[17,24]. This means that the number of mobile polarons $p$ can be smaller than that of ionized dopants $N_A^-$ for decreasing temperatures, i.e., instead of Eq. (1) the generalized neutrality condition

$$p + N_{CT}^+ = N_A^- \qquad (4)$$

should be applied for doped organics. Here, $N_{CT}^+$ denotes the number of holes provided by doping, but bound in [D$^+$A$^-$] ICTCs with binding energy $E_{CT}^b$. In a respective model for thermal activation of holes from energetically broadened density of ICTCs into free holes around energy $E_V$ (cf. Figure 1b, blue Gaussians), the population of holes bound in [D$^+$A$^-$] ICTCs follows

$$N_{CT}^+ = \int dE\, g_{CT}(N_A^-) \times (1 - f_{FD}(E_F, T)) \quad , \qquad (5)$$

with the ICTC density given by

$$g_{CT}(N_A^-) = \frac{N_A^-}{\sqrt{2\pi}\sigma_{CT}} \exp\left(-\frac{(E - E_{CT}^b)^2}{2\sigma_{CT}^2}\right) \quad , \qquad (6)$$

and $f_{FD}(E_F,T)$ as Fermi-Dirac statistics. With the generalized neutrality condition Eq. (4), it follows (cf. Supplementary Note 1, approximation of a $\delta$-type $E_{CT}^b$)

$$p(T, N_A) = \frac{N_A^-(T, N_A)}{1 + \exp\left(\frac{E_{CT}^b - E_F}{k_B T}\right)} \equiv N_{CT}^-(T) \quad , \qquad (7)$$

which is similar to Eq. (1), but here related to an occupation of $N_A^-$ [D$^+$A$^-$] ICTCs with electrons. Those electrons correspond to isolated acceptor anions A$^-$ and are, hence, denoted by $N_{CT}^-$. For all ionized acceptor molecules, the density condition

$$N_{CT}^+ + N_{CT}^- = N_A^- \qquad (8)$$

holds.

Solutions of Eq. (4) are shown in Fig. 4a–c for typical parameters ($E_{CT}^b = 0.64$ eV, $\sigma_{CT} = \sigma_V = 1...150$ meV). Despite complete ICT, $N_A^- = N_A$, a decreasing doping efficiency $p/N_A$ with rising $N_A$ is obtained due to enhanced population of ICTCs with holes, $N_{CT}^+$, reflecting experimental trends[9–11,30–34]. The Fermi level $E_F$ is pinned below $E_{CT}^b$, just as for the previously assumed $E_A$ in Eq. (1) (detailed comparison in Supplementary Figure 5 and 6). Varying temperature inverts the fraction of charges being free or bound in [D$^+$A$^-$] ICTCs, whereas $N_A^-(T)$ is constant such as seen in absorption (cf. Figure 2c,d). Without a broadening of the ICTC binding energy ($\sigma_{CT} = 1$ meV), the doping efficiency is negligibly low due to the strong Coulomb

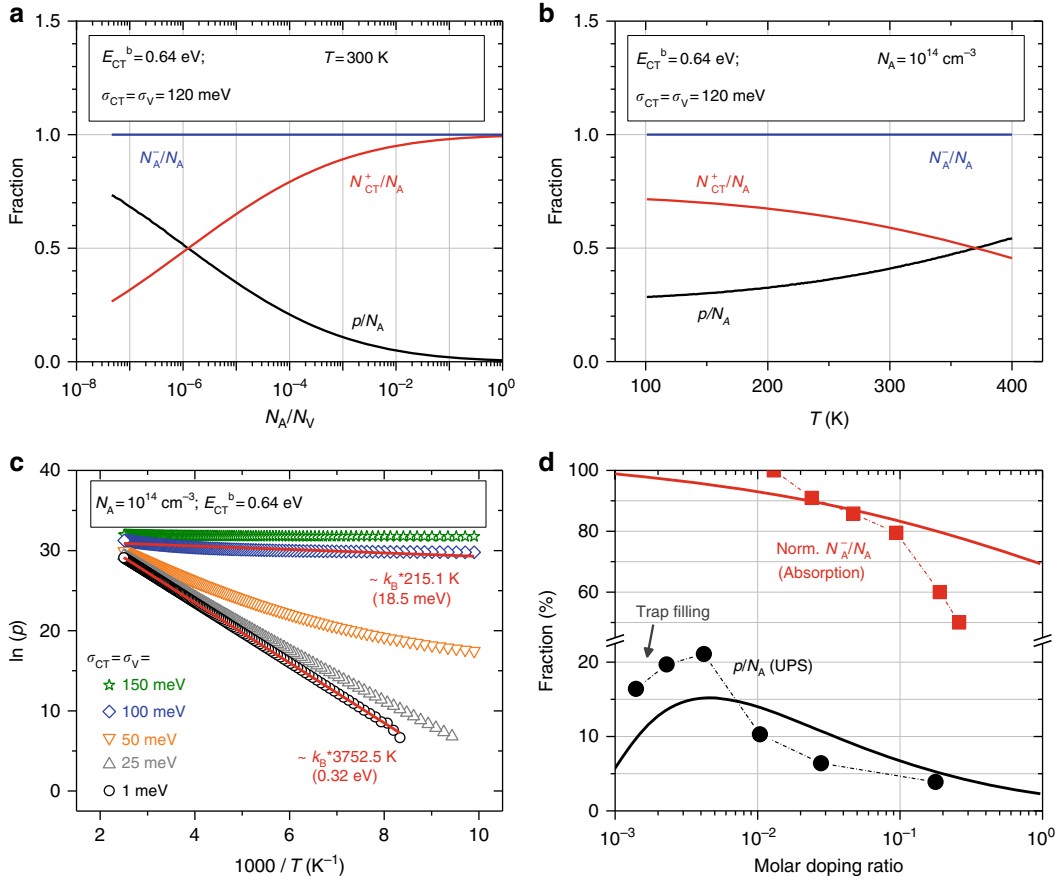

**Fig. 4** Linking doping freeze-out and ICTC occupation. **a–c** Numerical solutions of the neutrality condition Eq. (4) for $E_{CT}^b = 0.64$ eV and $N_V = 2.4 \times 10^{19}$ cm$^{-3}$. **d** Doping efficiencies $p/N_A$ determined by incremental UPS on Ag/MeO-TPD(0.7 nm)/ZnPc(4.5 nm)/ZnPc:F$_6$-TCNNQ($x$,MR) samples and fraction of ionized acceptors $N_A^-/N_A$ (F$_6$-TCNNQ$^-$/F$_6$-TCNNQ) estimated from the absorption spectra of Fig. 2. Continuous lines indicate calculations using a three-level model with $E_{CT}^b = 0.64$ eV, $E_A = 0.25$ eV, and $\sigma_{CT} = \sigma_A = \sigma_V = 160$ meV, considering trap filling[10]

binding of $E_{CT}^b = 0.64$ eV. In classical semiconductor theory, this situation corresponds to doping reserve, i.e., the presence of an equivalently deep dopant level $E_A$ and, indeed, the calculation yields a free hole activation of $(E_{CT}^b - E_V)/2 = 0.32$ eV as shown in Fig. 4c (see also Supplementary Figure 7 and 8). With increasing energetic disorder, however, the temperature-dependence $p(T)$ becomes significantly weakened, e.g., reaching $E_{act} \approx 19$ meV for a broadening of 100 meV, which compares with the values of Table 1 and is typical for small-molecule semiconductors[9,18,22,47,48]. We conclude that energetic disorder is essential for dissociation of such [D$^+$A$^-$] ICTCs. This also explains why the amorphous MeO-TPD yields systematically lower activation energies than the crystalline ZnPc or Pentacene (P5) (cf. Table 1) due to higher energetic disorder $\sigma$ as, e.g., seen in UPS (0.24 eV vs. 0.18 or 0.16 eV)[9,10,48]. This trend is further supported by the amorphous system Ir(piq)$_3$:F$_6$-TCNNQ, which shows a thermal activation of only 9.5 meV, i.e., as low as $E_{act}$ for MeO-TPD:F$_6$-TCNNQ (see Supplementary Figure 3 and Table 1).

In addition to (intrinsic) energetic disorder, the absolute dissociation probability is determined by electrostatic interactions in the specific D:A system. The ICTC binding energy $E_{CT}^b$ is governed by the strong Coulomb attraction between electron and hole, and further altered by interactions of the charges with quadrupole and induced dipole moments of the surrounding molecules[12]. The contribution due to Coulomb attraction becomes weaker as the dopant and host wavefunctions localize more strongly, and as the hole and electron become more distant. Thus, low activation energies are expected for large molecules

such as fullerenes. However, doping C$_{60}$F$_{36}$ rather than F$_6$-TCNNQ into the investigated hosts yields systematically higher activation energies reaching even 54 meV for P5:C$_{60}$F$_{36}$ (cf. Table 1). This observation rather points to electrostatic interactions of the [D$^+$A$^-$] ICTC with surrounding molecules that actually favor the ICTC dissociation, an effect previously estimated to 0.1–0.5 eV[12,49].

Electrostatic interactions were previously demonstrated to facilitate CT state dissociation in organic photovoltaic D:A blends[50], in particular for planar donor molecules with quadrupole moments such as P5[51,52]. In the idealized case of flat D/A interfaces with C$_{60}$ as quadrupole-free acceptor, respective charge–quadrupole interactions were demonstrated to yield either repulsive or attractive forces, depending on the inter-molecular orientation of the donor, i.e., its quadrupole moment, with respect to the acceptor[51,53]. Similarly, it is hence reasonable to presume that the flat F$_6$-TCNNQ dopant with its strong quadrupole moment reduces ICTC-binding energies compared with the more spherical C$_{60}$F$_{36}$ dopant and, indeed, $E_{CT}^b$ (estimated from $E_F$ pinning in UPS) is lower for ZnPc:F$_6$-TCNNQ (0.64 eV) by 0.24 eV than for P5:C$_{60}$F$_{36}$ (0.88 eV)[10]. As the intrinsic energetic disorder is similar for both hosts, we presume that the quadrupolar nature of the planar F$_6$-TCNNQ dopant is responsible for the systematically lower $E_{act}$ compared with C$_{60}$F$_{36}$: 19 meV vs. 54 meV (P5:F$_6$-TCNNQ vs. P5:C$_{60}$F$_{36}$) and 9.1 meV vs. 16 meV (MeO-TPD:F$_6$-TCNNQ vs. MeO-TPD:C$_{60}$F$_{36}$). Thus, this direct comparison provides a crucial design rule for efficient host:dopant systems.

**Doping reserve at high concentrations.** The suppressed capacitive response at high doping ratios in impedance spectroscopy prevents unambiguous proof of the reserve regime for high doping concentrations relevant for practical applications. We therefore perform UPS on metal/organic interfaces with gradually increasing thicknesses, providing evidence for the reserve regime for ZnPc:F$_6$-TCNNQ. The method and analysis are similar to those previously used (cf. refs. [9,30] and Supplementary Note 2). The resolved depletion layer widths $w$ at RT are on the order of 3… 23 nm (Supplementary Figures 10 and 11, and Supplementary Table 1). The determined ratio $p/N_A$ is plotted in Fig. 4d, possessing a maximum of ~ 21% at MR = 0.004 and dropping just below 4% at MR = 0.18. Compared with the fraction of ionized dopant molecules $N_A^-/N_A$ estimated from the intensity of the F$_6$-TCNNQ$^-$ absorption features (cf. Supplementary Figure 2), a monotonously decreasing trend is found. However, the ratio at MR = 0.1 is yet 80% of that at MR = 0.01, i.e., it shows a much weaker relative drop than $p/N_A$ (see Fig. 4d). We thus conclude that, first, the doping efficiency indeed drops due to enhanced binding of holes in [D$^+$A$^-$] ICTCs and, second, 100% dopant ionization is not sufficient to precisely describe the doping process for ZnPc:F$_6$-TCNNQ. Essentially, the whole mechanism can be fully understood in terms of a three-level model, in which acceptor states $E_A < E_{CT}^b$ determine dopant ionization $N_A^-$ in Eq. (4) in equilibrium (cf. Figure 1b). The dependency $p(T)$ is yet mostly decoupled from $N_A^-(T)$, as holes are first bound in ICTCs of higher binding energy before being thermally activated upon dissociation. The situation of weak dopants, i.e., systems with $EA$ $(A) < IE(D)$, is covered by rather deep acceptor states (cf. Supplementary Figure 9). Identifying $E_A$ for ZnPc:F$_6$-TCNNQ is not unambiguous, as the estimation for $N_A^-/N_A$ allows only for conclusions on trends rather than absolute values due to unknown absorption cross-sections. Nonetheless, a possible calculation ($E_A = 0.25$ eV, $E_{CT}^b = 0.64$ eV), shown in Fig. 4d, approximates the measured doping efficiency well, including the drop due to trap filling at concentrations below 0.004[10].

Doping freeze-out is further confirmed by UPS on several Ag/ZnPc:F$_6$-TCNNQ($x_m$) samples under varying temperature (93 K < $T$ < 300 K). Under freeze-out (reserve) conditions, $T$-dependent depletion widths $w(T)$ are expected. UPS spectra and Fermi level positions are given in Supplementary Figure 12 and are compared with calculated level bendings in Supplementary Figure 11, taking the 20.7 meV Arrhenius-type activation into account (details given in Supplementary Note 3). For samples where $w(T)$ extends beyond the organic layer thickness $x_m$ upon cooling, a consistent shift of $E_F(T)$ toward mid-gap by several 100 meV is found in UPS, e.g., for MR = 0.01 where $w(T = $ RT$) \approx 14$ nm < $x_m = 16$ nm < $w(T = 100$ K$) = 33.2$ nm (cf. Figure 3d). For doping ratios as low as 0.002, $E_F(T)$ measured by UPS even reaches its intrinsic position at 100 K. The carrier density provided by doping is not sufficient to fill the deep traps anymore. Thus, the depletion width rapidly extends beyond 100 nm as in an undoped sample where the level alignment is governed by interface effects[54,55].

**Simulating the conductivity scaling.** The conductivity was identified to super-linearly scale with the dopant concentration for ZnPc:F$_6$-TCNNQ[10], despite the evidence for the reserve regime. To clarify this discrepancy, we here provide kinetic Monte Carlo (KMC) transport simulations, taking particularly mutual electrostatic Coulomb interactions of ionized dopants and transferred charge carriers into account. Boxes with up to 51$^3$ sites of Gaussian disordered energies (DOS) and Miller–Abraham hopping rates are used for determining the conductivity (for details, see Supplementary Note 4). As shown in Fig. 5a, the measured log–log scaling can be well reproduced (simulated slope ~1.64), including the effect of trap filling characterized by an even

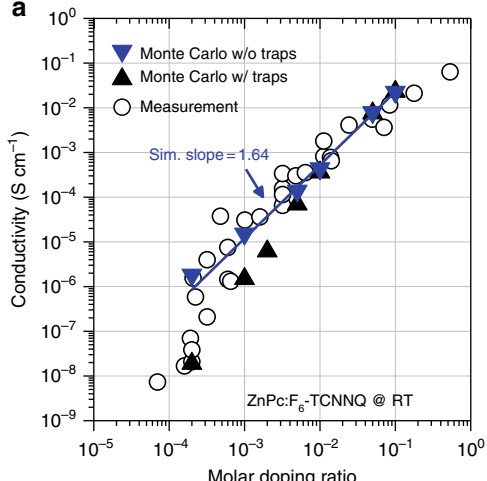

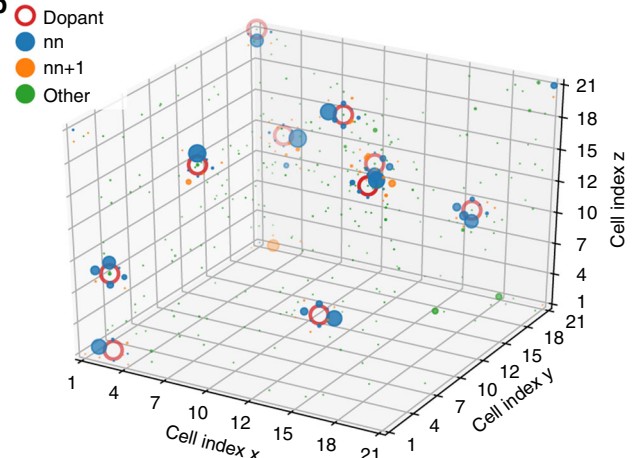

**Fig. 5** Monte Carlo simulation of the conductivity. **a** Super-linear scaling of the conductivity vs. doping ratio for ZnPc:F$_6$-TCNNQ at RT. Experimental values are taken from ref. [10], the simulation yields a log–log slope ~1.64. At low doping ratios, deep traps with $N_T = 7.2 \times 10^{17}$ cm$^{-3}$ are filled. **b** Illustration of the simulated occupation probability $W$ of sites with charge carriers (holes) in a 21 × 21 × 21 mesh with periodic boundary conditions and MR = 10$^{-3}$ (10 dopants, without traps). Blue and orange dots represent the occupation probabilities of nearest-neighbor (nn) and next nearest-neighbor (nn + 1) sites of dopants, respectively, which positions are marked as red circles. For all other sites, the occupation probability is indicated by green dots. The apparently isolated orange dot at site (2,21,1) is associated to the dopant at site (1,21,21), i.e., its equivalent in the bottom periodic replica of the shown mesh. The occupation probability $W$ increases with the plotted dot size each, here shown for a threshold of $W > 10^{-4}$. Dots located at positions farther away from the viewing angle are plotted with higher transparency. A respective site occupation plot considering deep trap states is given in Supplementary Figure 13

steeper scaling at concentrations below 0.005. KMC further allows to analyze the actual spatial distribution of the carriers, which is shown in Fig. 5b and Supplementary Movie 1. On average, most of the carriers are found on nearest-neighbor (nn) or next nearest-neighbor (nn + 1) sites of ionized dopants, comprising > 80% for MR > 0.01 (cf. Supplementary Figure 13). This resembles the trends in $N_{CT}^+$ determined by Eq. (4) and UPS (cf. Figure 4); similar trends were recently published by the Kemerink group[25]. Hence, we conclude that those carriers bound in ICTCs should partially contribute to the super-linearly rising conductivity, in particular at high doping ratios at which overlapping Coulomb potentials render them increasingly mobile in

addition to the already free carriers $p$. Thus, the mobility effectively increases (cf. Supplementary Figure 13). At doping ratios above 0.1, the fraction of free $p$ is only a few percent (strong reserve, cf. Figure 4); hence, the transport should be significantly impacted by carrier hopping between bound $[D^+A^-]$ ICTCs.

These findings clearly open up a roadmap for future research and for designing novel host:dopant systems, i.e., exactly investigating the electrostatic interactions of D:A systems on a molecular level and determining energetic disorder with an accuracy of ideally below 10 meV, which is both experimentally and theoretically challenging[38,48,56].

## Discussion

This work demonstrates that molecular doping is a two-step process, comprising single-electron transfer from donor to acceptor molecules, and subsequent dissociation of formed $[D^+A^-]$ ICTCs, which determine the overall doping efficiency in excitonic organic semiconductors. In particular, we show that ICTC dissociation resulting in mobile polarons is of Arrhenius-type with a thermal activation of only a few 10 meV, although the Coulomb binding energy $E_{CT}^b$ of charges in ICTCs typically comprises several 100 meV. This effective lowering is understood to originate from energetic disorder, hence being essential for the functioning of molecular doping in general. The concentration and temperature dependence of the doping efficiency is well explained in framework of an extended semiconductor statistics description in which ICTC occupation causes the classically known reserve regime even for 100% dopant ionization. The border cases of weak molecular dopants, i.e., the presence of deep acceptor states $E_A$ for p-doping, as well as classical semiconductor physics are included, namely for decreasing ICTC binding energy $E_{CT}^b \to 0$ when $E_{CT} \leq E_A$ is reached. With this, a complete description of the mechanism and statistics of molecular DA doping is established.

## Methods

**Materials/preparation**. The doped layers are thermally co-evaporated at RT under ultra-high vacuum (UHV) conditions (base pressure $1 \times 10^{-9}$ mbar) by controlling the evaporation rates with two independent quartz crystal microbalances (QCMs). Molar doping ratios below 0.005 are achieved by evaporating the dopant molecules through a rotating shutter (2… 3 Hz) positioned between the substrate and the QCM. The circular shutter is partly opened ($\simeq 18°$), which reduces the effective molecule transmission to ~ 5%. The host materials zinc-phthalocyanine (ZnPc, CreaPhys, Dresden, Germany), MeO-TPD (Sensient, Wolfen, Germany), and Tris(1-phenylisoquinoline) iridium(III) (Ir(pic)₃, American Dye Source, Inc., Baie-D'Urfé, Canada) were purified at least twice by a three-zone vacuum gradient sublimation. The dopant compound $F_6$-TCNNQ was purchased from Novaled GmbH (Dresden, Germany) and was used as delivered. Chemical structures are shown in Fig. 2 of the main article.

**Absorption**. Absolute absorption measurements were carried out using the integration sphere unit of an UV-Vis-NIR photospectrometer (SolidSpec-3700, Shimadzu, Japan). The integrated transmission and reflection were measured for films deposited on glass.

**Temperature-dependent transmission**. The white light of a 50 W halogen lamp is focused onto the organic film, deposited onto a glass substrate, chopped at a frequency of 141 Hz, and afterwards coupled into a monochromator (Cornerstone 260 1/4 m, Newport). The resulting mono-chromatic light is detected with a calibrated indium–gallium–arsenide photodiode, its current is fed into current–voltage pre-amplifier, and analyzed with a lock-in amplifier (Signal Recovery 7280 DSP, USA). The transmission is determined by comparing the substrate with the organic film and a neat substrate. The sample is cooled down to 10 K by keeping it in a helium vapor in a continuous flow cryostat (STVP-100, Janis Research, USA).

**Impedence spectroscopy**. Impedence spectroscopy was performed with an Autolab PGSTAT302N LCR in the range from 1 MHz down to 0.1 Hz. The sample was kept at zero bias, whereas the excitation signal amplitude was at 15 mV. All measurements were performed in the dark utilizing either a continuous He vapor flow cryostat (STVP-100, Janis Research) for temperature variation down to 20 K or by using a Peltier element. The capacitance function is calculated

from the imaginary part of the admittance $Y$ assuming an RC-equivalent circuit: $C(f) = \text{Im}[Y(f)/f]$.

For Mott–Schottky analysis, measurement conditions that allow a determination of the active dopant density/activation were chosen for each material, i.e., the following two criteria are guaranteed: (1) sufficient capacitive behavior of the device, i.e., ensuring measurement of the actual depletion capacitance $C_d$ at the metal/semiconductor Schottky contact being neither superimposed by carrier injection when varying the voltage in $C(V)$ nor by the geometric capacitance of the organic film. Both aspects are achieved by choosing neither too high nor too low doping ratios. (2) Avoiding capacity contributions originating from charging and discharging of trap states, which are characterized by different slopes in $C^{-2}(V)$ Mott–Schottky plots, i.e., causing kinks in those. Accordingly, bias voltages down to $-1.0$ V (p-MeO-TPD), $-2.0$ V (p-Ir(piq)₃), and $-0.2$ V $< V < 0.3$ V (p-ZnPc) were used as analysis ranges. For the latter, Mott–Schottky analysis was performed on devices of varying doping ratio and organic layer thickness as control experiments (Supplementary Figure 3 and 4), from which an uncertainty of 5 meV for the Arrhenius-type activation energy $E_{act}$ was estimated.

**Ultraviolet photoelectron spectroscopy**. The UPS measurements are performed with a Phoibos 100 system (Specs, Berlin, Germany) under UHV conditions (base pressure $5 \times 10^{-11}$ mbar) and sputter cleaned silver foil (99.995%, MaTecK, Juelich, Germany) is used as substrate. Sample transfer without breaking vacuum conditions is ensured by a direct connection of the UPS to the evaporation chamber. The energy resolution (HeI, 21.22 eV) is 130 meV and the reproducibility is estimated to 50 meV. During the UPS measurement the sample is set to an acceleration potential of $-8$ V. For each spectrum, the emission features due to secondary line excitations of the HeI discharge lamp are subtracted. The measurements are kept as short as possible to avoid degradation of the organic materials and charging effects. The samples were cooled by a flow of liquid nitrogen trough the substrate holder of the UPS analysis chamber, recording the temperature by thermocouple positioned on the metallic holder close to the actual sample under investigation.

**Data availability**. All data that support the findings of this study are available from the corresponding authors upon reasonable request.

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

## Acknowledgements

This research was funded by the German Federal Ministry for Education and Research (BMBF) through the InnoProfile project "Organische p-i-n Bauelemente 2.2," as well as competitive funding from the King Abdullah University of Science and Technology. In addition, this work received funding from the European Union Seventh Framework Programme under the grant agreement number 607232 (THINFACE), from the Austrian Science Fund (FWF), grant I2081-N20, and finally from the German Research Foundation (DFG) through the project MatWorldNet LE-747/44-1. We thank Professor Björn Lüssem and Dr Christian Körner for fruitful discussions. K.L. thanks the Canadian Institute for Advanced Research (CIFAR) for support.

## Author contributions

M.T. motivated and conceived the project, wrote the manuscript, developed the doping model, did numerical calculations, and performed RT and T-dependent UPS measurements. J.B. and M.T. carried out absorption/transmission measurements. P.P., M.T., and H.K. performed impedance spectroscopy and Mott–Schottky analysis. B.N., M.K, and K.Z. provided the Monte Carlo simulations. M.S. contributed with T-dependent UPS measurements. K.V. and K.L. supervised the work and contributed with fruitful discussions.

## Additional information

**Competing interests:** The authors declare no competing interests.

