## [Peer Review File · Nature Communications]

Reviewers' comments:

Reviewer #1 (Remarks to the Author):

In this work, Tietze et al. provided insight into the fundamental mechanisms of doping in organic semiconductors, for which a consolidated model is still lacking in this field of organic electronics. The authors presented a set of experimental results corroborated by modeling employing ZnPc:F6-TCNNQ and MeO-TPD:F6-TCNNQ materials as model systems for studying the p-doping behavior. In their proposed model, the doping is based on two-step process: (i) single electron transfer from donor to acceptor and (ii) separation of formed ground-state integer-charge-transfer complex (ICTC), Figure 1a. A central question that the authors address in this work is the fundamental origin for the additional energy (or activation energy, E_a) that has to be supplied to the system in order to generate a free hole (or polaron in organic films) in ZnPc and MeO-TPD host molecules. Otherwise, spontaneous doping would occur based solely on energy levels of (properly selected) host and dopant material systems. They performed (1) temperature-dependent UV-vis-NIR evidencing that ICTC formation is temperature-independent occurring even at a low temperature 10 K (i.e., ICT depends only on the energy levels of donor (D) and acceptor (A): $IE(D) < EA(A)$). (2) temperature-dependent impedance spectroscopy (Mott-Schottky analysis) on ITO/(ZnPc or MeO-TPD):F6-TCNNQ(50 nm)/Al devices revealed activation energies for hole release of ~ 9 meV and ~ 21 meV in ZnPc:F6-TCNNQ and MeO-TPD:F6-TCNNQ, respectively. (3) Corroborated by numerical solutions of charge neutrality conditions (page 8), the authors concluded that energetic disorder is key to dissociation of ICTC. In addition, it was shown that the amorphous nature of MeO-TPD leads to a lower activation energy of ~ 9 meV for ICTC dissociation. (4) The freeze-out (reserve) conditions in ZnPc:F6-TCNNQ and MeO-TPD:F6-TCNNQ material systems were demonstrated based on the depletion widths at metal/organic contacts measured by ultraviolet photoemission spectroscopy (doping concentrations and temperatures). (5) Monte-Carlo simulation was employed to explain the discrepancy in the freeze-out condition and linearity in conductivity as a function of dopant concentration. The manuscript is well organized with proper flow for developing the authors' proposed model for doping in organic semiconductors. The previous works were presented properly in the introduction and discussion sections. Because of its high quality and value to the field of organic electronics, this work is suitable for publication in Nature Communications after the authors address the following minor points.

(1) On page 4, line 75: It is better to specify that the 9 meV activation energy for hole release corresponds to the MeO-TPD:F6-TCNNQ system.

(2) At the end of section "Temperature-independent polaron absorption" (page 6), can the authors add a brief conclusion or extend the interpretation for "no remarkable intensity drop is noticeable for F6-TCNNQ in both hosts..."? Do you mean that the ICT process is only energy level dependent (ionization energy of the host material and the electron affinity of the p-dopant)? Please extend the discussion.

(3) On page 9, line 183, the authors concluded that energetic disorder is key to inducing ICTC dissociation. This conclusion seems reasonable, but it will be helpful if the authors can provide some supporting references. If this has not been studied previously, would it be possible to perform a control experiment using the same host:dopant material system and varying the crystallinity of the material system to study the behavior of activation energies for hole release? A direct correlation of crystallinity / energetic disorder versus activation energy would be strong evidence for the authors' proposed model.

Reviewer #2 (Remarks to the Author):

This paper investigates the process of charge transfer doping in molecular semiconductors, in particular the generation of free carriers that requires overcoming the strong Coulomb interaction between the carriers. It is claimed that a surprisingly low activation energy for generation of free carriers is a manifestation of the energetic disorder of the molecular semiconductor.

The scientific insights of the paper are interesting, though the insight gained are in my view not fundamentally novel. The understanding of doping as a two-step process and the role of energetic disorder in facilitating carrier dissociation has been clearly established previously (see for example Ref 12). The value of the paper consists mainly in establishing this in a wider range of materials' systems as well as in a description of the relevant physics in terms of classical semiconductor physics.

My main comment is that some aspects of the paper could be presented more clearly: I found the diagram in Figure 1b rather confusing. There are coloured and black density of states distributions that are not clearly explained and also it is not clear what determines exactly the activation energy. What points in the tail determine the activation energy? What impact does the inclusion of the three-state model have on the activation energy? It might also be helpful to include a diagram to illustrate the reserve regime within the three-state model.

I was also confused by Figure 5b and probably do not understand clearly what this plot represents. I see for example one orange dot in the centre of the plot, which should be a $nn+1$ carrier but it is nowhere near a dopant. There is also a fine "mist" of dots scattered all over the plot. I presume these are all green and represent free carriers?

I also have questions about the C-V plots in Figure S3. They exhibit clear kinks and the choice of regime that was used to extract dopant density appears somewhat arbitrary. Certainly it appears that for ZnPc a low voltage regime was chosen for the fits while for MeO-TPD a higher voltage regime was selected. This needs to be explained and justified.

On page 9 of the main text the authors argue that the higher activation energies obtained for C60F36 is a reflection of a less favourable electrostatics. I found this somewhat surprising on the basis that for the fullerene dopant the electron wavefunction is less localised than for the F6-TCNNQ, i.e. one might have expected the Coulomb binding energy to be lower because of a larger distance between electron and hole? Why does this not manifest itself?

Reviewer #3 (Remarks to the Author):

In their article „Elementary steps in electrical doping of organic semiconductors“, M.L. Tietze et al. address the process of charge separation on molecularly doped organic semiconductors. While the process of charge transfer has been elucidated during the past years showing two competing processes of ground-state integer charge transfer (ICT) and host-dopant electronic wave-function hybridization, the subsequent step of carrier release, key for the generation of mobile electrons/holes upon n-/p-doping organic semiconductors, is still not fully understood today. In particular, it is not fully resolved how electron and hole after initial charge transfer can overcome their substantial Coulomb binding energy, i.e., how the thusly-formed integer charge transfer complexes (ICTCs) can thermally separate, as Coulomb binding contributes significantly more than kT can provide at room temperature.

The authors thoroughly address this important question both experimentally and by theoretical modelling. From temperature-dependent absorption spectroscopy on F6TCNNQ-doped ZnPc and MeO-TPD films, two organic semiconductors with ionization energies lower than the electron affinity of the chosen dopant, they conclude that ICT does not significantly depend on the temperature down to 10K, i.e., doping freeze-out is not due to less ionized acceptor molecules. From impedance spectroscopy (in a safe temperature range where deep traps are filled) they deduce Arrhenius-type activation energies of 20 meV and below, in particular, significantly lower than the expected Coulomb binding energy of an ICTC. The authors further stress that MeO-TPD, suggested to form amorphous films, has an even lower value than expectedly crystalline ZnPc, while the temperature dependence of the relative density of ionized acceptors is stronger for the latter. This fits well to the outcome of their modelling of ICTC occupation based on extensive (temperature-dependent) UPS data, where they conclude that energetic disorder is key for the dissociation of the ICTC, as it effectively lowers the activation energy. By their UPS analysis, the authors further provide evidence for temperature-dependent depletion widths, which are indicative for doping freeze-out.

Overall, to my point of view the authors provide a convincing model for mobile charge generation after the initial processes of ICTC formation. Their model is based on a solid footing of experimental data and the theoretical treatment thereof and will valuably contribute to a timely topic in the field of organic electronics. The data is well documented and the experimental information provided seems sufficient to me to reproduce the data. I would suggest publishing the present manuscript as it is.

We thank the three reviewers for their very helpful comments. In the following, we reply to the comments and explain the changes made in the manuscript.

REVIEWER #1:

In this work, Tietze et al. provided insight into the fundamental mechanisms of doping in organic semiconductors, for which a consolidated model is still lacking in this field of organic electronics. The authors presented a set of experimental results corroborated by modeling employing ZnPc:F₆-TCNNQ and MeO-TPD:F₆-TCNNQ materials as model systems for studying the p-doping behavior. In their proposed model, the doping is based on two-step process: (i) single electron transfer from donor to acceptor and (ii) separation of formed ground-state integer-charge-transfer complex (ICTC), Figure 1a. A central question that the authors address in this work is the fundamental origin for the additional energy (or activation energy, E_a) that has to be supplied to the system in order to generate a free hole (or polaron in organic films) in ZnPc and MeO-TPD host molecules. Otherwise, spontaneous doping would occur based solely on energy levels of (properly selected) host and dopant material systems. They performed (1) temperature-dependent UV-vis-NIR evidencing that ICTC formation is temperature-independent occurring even at a low temperature 10 K (i.e., ICT depends only on the energy levels of donor (D) and acceptor (A): IE(D) < EA(A)). (2) temperature-dependent impedance spectroscopy (Mott-Schottky analysis) on ITO/(ZnPc or MeO-TPD):F₆-TCNNQ(50 nm)/Al devices revealed activation energies for hole release of ~9 meV and ~21 meV in ZnPc:F₆-TCNNQ and MeO-TPD:F₆-TCNNQ, respectively. (3) Corroborated by numerical solutions of charge neutrality conditions (page 8), the authors concluded that energetic disorder is key to dissociation of ICTC. In addition, it was shown that the amorphous nature of MeO-TPD leads to a lower activation energy of ~9 meV for ICTC dissociation. (4) The freeze-out (reserve) conditions in ZnPc:F₆-TCNNQ and MeO-TPD:F₆-TCNNQ material systems were demonstrated based on the depletion widths at metal/organic contacts measured by ultraviolet photoemission spectroscopy (doping concentrations and temperatures). (5) Monte-Carlo simulation was employed to explain the discrepancy in the freeze-out condition and linearity in conductivity as a function of dopant concentration.

The manuscript is well organized with proper flow for developing the authors' proposed model for doping in organic semiconductors. The works were presented properly in the introduction and discussion sections. Because of its high quality and value to the field of organic electronics, this work is suitable for publication in Nature Communications after the authors address the following minor points.

(1) On page 4, line 75: It is better to specify that the 9 meV activation energy for hole release corresponds to the MeO-TPD:F₆-TCNNQ system.

Reply: As suggested by the reviewer, we specifically point out in the introduction that this exceptionally low value corresponds to the MeO-TPD:F₆-TCNNQ system.

Text: “[...] activation for hole release determined by Mott-Schottky analysis is only a few 10 meV and can be even such low as 9 meV as obtained for the

prototypical OLED hole transporter system MeO-TPD:F₆-TCNNQ. This effective lowering is ascribed to originate from energetic disorder. [...]"

(2) At the end of section "Temperature-independent polaron absorption" (page 6), can the authors add a brief conclusion or extend the interpretation for "no remarkable intensity drop is noticeable for F₆-TCNNQ in both hosts..."? Do you mean that the ICT process is only energy level dependent (ionization energy of the host material and the electron affinity of the p-dopant)? Please extend the discussion.

Reply: We slightly extended the conclusion as requested by the reviewer. We added the following:

Text: "Therefore, we conclude that the degree of dopant ionization, i.e, host-dopant ICT, is not (or negligibly weak) temperature-activated, in particular, for D:A systems with $EA(A) > IE(D)$ as studied here."

(3) On page 9, line 183, the authors concluded that energetic disorder is key to inducing ICTC dissociation. This conclusion seems reasonable, but it will be helpful if the authors can provide some supporting references. If this has not been studied previously, would it be possible to perform a control experiment using the same host:dopant material system and varying the crystallinity of the material system to study the behavior of activation energies for hole release? A direct correlation of crystallinity / energetic disorder versus activation energy would be strong evidence for the authors' proposed model.

Reply: We are glad that the reviewer agrees with our conclusion. A comparison of activation energies including previously determined values for the crystalline material pentacene (P5) p-doped by F₆-TCNNQ or C₆₀F₃₆ (Ref. 18) is given in Table 1. Here, the crystalline materials possess similar energetic disorders which are 0.18 eV (ZnPc) and 0.16 eV (P5), respectively, and are lower than that of the amorphous MeO-TPD (0.24 eV) as measured by UPS (see Refs. 9,10,47). Accordingly, the activation energy is found to be lower for the amorphous MeO-TPD regardless of the used dopant. To support this trend, we further provide Mott-Schottky data on the amorphous OLED material Ir(piq)₃ p-doped by F₆-TCNNQ, yielding a thermal activation of only 9.5 meV which is similar to the value obtained for MeO-TPD:F₆-TCNNQ. Accordingly, we updated Table 1, added the measurement data to Supplementary Fig. 3, and slightly extended the discussion on pg. 9:

Text: "[...] This trend is further supported by the amorphous system Ir(piq)₃:F₆-TCNNQ which shows a thermal activation of only 9.5 meV, i.e., as low as E_{act} for MeO-TPD:F₆-TCNNQ (see Supplementary Figure 3 and Table 1)."

	F ₆ -TCNNQ	C ₆₀ F ₃₆
MeO-TPD	9.1	16 ^[9]
Ir(piq) ₃	9.5	
Pentacene (P5)	19 ^[18]	54 ^[18]
ZnPc	21	

We believe this dataset is sufficient to argue that energetic disorder is key for ICTC dissociation.

Nevertheless, we agree with the reviewer that manipulating/correlating crystallinity, disorder, and doping activation for the same material would be an interesting topic, which however, should be studied in a separate work in detail. Doping-induced energetic disorder (see, e.g., Ref. 20) as well as doping-induced crystal growth suppression (see, e.g., 10.1016/j.orgel.2011.09.027) will render such a study experimentally challenging and extensive, hence, might blow up the present work without adding significant conclusions.

The Impedance measurements on $\text{Ir}(\text{piq})_3\text{:F}_6\text{-TCNNQ}$ were performed by Hans Kleemann, therefore, we added him as co-author and removed him from the Acknowledgments. Also, we updated the Experimental section.

REVIEWER #2:

This paper investigates the process of charge transfer doping in molecular semiconductors, in particular the generation of free carriers that requires overcoming the strong Coulomb interaction between the carriers. It is claimed that a surprisingly low activation energy for generation of free carriers is a manifestation of the energetic disorder of the molecular semiconductor.

The scientific insights of the paper are interesting, though the insight gained are in my view not fundamentally novel. The understanding of doping as a two-step process and the role of energetic disorder in facilitating carrier dissociation has been clearly established previously (see for example Ref. 12). The value of the paper consists mainly in establishing this in a wider range of materials' systems as well as in a description of the relevant physics in terms of classical semiconductor physics.

My main comment is that some aspects of the paper could be presented more clearly:

(1) I found the diagram in Figure 1b rather confusing. There are coloured and black density of states distributions that are not clearly explained and also it is not clear what determines exactly the activation energy. What points in the tail determine the activation energy? What impact does the inclusion of the three-state model have on the activation energy? It might also be helpful to include a diagram to illustrate the reserve regime within the three-state model.

Reply: We thank the reviewer for the comment. Accordingly, we modified Fig.

1b and its caption to make the diagram more clear to the reader.

Occupation of the acceptor level E_A with electrons has been changed to a gray distribution for better distinction from occupation of the levels E_V and E_{CT}^b with holes which are given in black. The activation energy E_{act} can only effectively be marked in the diagram as its actual value is determined by solving the integral equation Eq. (5) for given energetic disorder and varying temperature, yielding correlations $p(T)$ as plotted in Fig. 4c. From those, the Arrhenius-type activation energy is fitted. Therefore, attributing E_{act} to certain (specific) points of the tails of E_{CT}^b and E_V is non-trivial. To avoid confusion, we replaced the dashed tail markings by a red arrow labeled with E_{act} to indicate that the thermal activation is significantly lowered as compared to the $E_{CT}^b - E_V$ difference due to energetic disorder. The extended caption of Fig. 1b reads:

Text: “[...] Black and gray distributions indicate the occupations of Gaussian broadened levels E_V or E_{CT}^b (each blue) with holes and acceptor states E_A (yellow) with electrons, respectively. The energy required for ICTC dissociation is reduced from $E_{CT}^b - E_V$ to E_{act} due to energetic disorder, i.e., from several 100 mV to effectively only a few tens of meV. ICT itself is temperature-independent. Incomplete ICT occurs if $IE(D) > EA(A)$, which is expressed by E_A describing the degree of acceptor ionization in equilibrium.”

In a three-state model the acceptor level E_A is included to consider an incomplete degree of dopant ionization as necessary for systems with $EA(A) < IE(D)$. This situation is equal to classical deep acceptor states, in which dopant ionization, N_A^- , is temperature-dependent. Thus, as shown in Supplementary Fig. 9, the thermal activation of the free carrier density p is enhanced since affected by both thermal release from ICTCs and thermally-activated dopant ionization. On the example of an energetic disorder of 100 meV, this effect is found to particularly become significant for $E_A - E_V > 0.3$ eV.

The transition from doping saturation to reserve in a three-state model is defined by the deeper level, i.e., it is either controlled by the ICTC Coulomb binding E_{CT}^b or acceptor activation E_A . For real organic systems, however, E_{CT}^b is the predominant factor as illustrated in Supplementary Fig. 9 ($E_{CT}^b = 0.64$ eV, $E_A = 0.1 \dots 0.5$ eV, $\sigma_V = \sigma_{CT} = \sigma_A = 100$ meV). For better illustration, we added plots of the Fermi level position vs. the doping ratio and temperature (see below) to Supplementary Fig. 9, showing that for device relevant doping concentrations and typical temperatures E_F is below E_{CT}^b (doping reserve) and slightly shifts towards mid-gap when increasing the acceptor state E_A .

(2) I was also confused by Figure 5b and probably do not understand clearly what this plot represents. I see for example one orange dot in the centre of the plot, which should be a $nn+1$ carrier but it is nowhere near a dopant. There is also a fine "mist" of dots scattered all over the plot. I presume these are all green and represent free carriers?

Reply: We thank the reviewer for his/her concerns regarding readability of Fig. 5b. The plot basically shows the timely-averaged occupation probability W of sites with holes in a $21 \times 21 \times 21$ mesh containing 10 dopants (marked by red circles), i.e., representing a molar doping ratio of 0.001. Here, blue and orange dots indicate (nn) and $(nn+1)$ positions to the dopants, respectively. The dot size scales with the occupation probability, i.e., large dots indicate high occupation probabilities on the average. Dots located at positions farther away from the viewing angle are plotted with higher transparency.

The orange dot in the middle indeed shows the hole occupation probability of a site which is a $(nn+1)$ position to the dopant being located at coordinates (1,21,21). It just appears at the other corner of the box due to periodic boundary conditions.

Yes, the reviewer is right, the fine "mist" of green dots scattered all over the plot represents occupation of sites, which are neither nearest neighbor (nn) nor next nearest-neighbor $(nn+1)$ sites of dopants, with charge carriers. In contrast to the blue and orange indicated (nn) and $(nn+1)$ sites, their occupation probability is rather low and just above the chosen plot threshold of $W=10^{-4}$. Therefore, carriers occupying the green sites can be considered as statistically free on the average. However, if including deep trap states, the occupation probability of corresponding trap sites is strongly enhanced. This can be seen in Supplementary Fig. 13c on the example of 6 trap and 10 dopant sites in the $21 \times 21 \times 21$ grid (representing a relative trap density of 0.0007 and doping ratio of 0.001), where the high trap occupation probability is consistently indicated by 6 large green dots. The previously obtained fine "mist" of green dots is diluted since the number of free carriers is reduced due to the relatively large trap density. For direct comparison, both cases are shown below:

For improving readability and clarity of Fig. 5b, we extended its caption. Furthermore, we provide as SI a gif-file, visualizing the simulated 3D mesh by rotating the viewing angle.

Text: “b Illustration of the simulated occupation probability W of sites with charge carriers (holes) in a $21 \times 21 \times 21$ mesh with periodic boundary conditions and $MR=10^{-3}$ (10 dopants, w/o traps). Blue and orange dots represent the occupation probabilities of nearest neighbor (nn) and next nearest-neighbor (nn+1) sites of dopants, respectively, which positions are marked as red circles. For all other sites, the occupation probability is indicated by green dots. The apparently isolated orange dot at site (2,21,1) is associated to the dopant at site (1,21,21), i.e., its equivalent in the bottom periodic replica of the shown mesh. The occupation probability W increases with the plotted dot size each, here shown for a threshold of $W > 10^{-4}$. Dots located at positions farther away from the viewing angle are plotted with higher transparency. A respective site occupation plot considering deep trap states is given in Supplementary Figure 13.”

(3) I also have questions about the C-V plots in Figure S3. They exhibit clear kinks and the choice of regime that was used to extract dopant density appears somewhat arbitrary. Certainly, it appears that for ZnPc a low voltage regime was chosen for the fits while for MeO-TPD a higher voltage regime was selected. This needs to be explained and justified.

Reply: The reviewer is right, the used regimes in Mott-Schottky analysis are different for MeO-TPD and ZnPc, however, have not been chosen arbitrary. When performing this analysis, two factors must be guaranteed to indeed determine the active dopant density/activation:

- (1) Sufficient capacitive behavior of the device, i.e., ensuring measurement of the actual depletion capacitance C_d at the metal/semiconductor Schottky contact being neither superimposed by carrier injection when varying the voltage in C-V nor by the geometric capacitance of the organic film. Both aspects are achieved by choosing neither too high nor too low doping ratios.
- (2) Avoiding capacity contributions originating from charging and de-charging of trap states, which are characterized by different slopes in $1/C^2(V)$ Mott-Schottky plots, i.e., causing kinks in those.

In case of MeO-TPD both criteria are fulfilled over a rather large voltage range down to -1.0 V. For ZnPc, however, the $1/C^2(V)$ plots show carrier injection in both forward and backward direction ($1/C^2$ values increase), particularly at the higher temperatures. Furthermore, kinks appear indicating trap response. Therefore, the

voltage range suitable for determining the $d(1/C^2)/dV$ slope attributable to the actual depletion capacitance is rather limited. With decreasing temperature, it also shifts to slight forward biases, ensuring trap saturation.

These circumstances complicate Mott-Schottky analysis for p-ZnPc, why we used $-0.2 \text{ V} < V < 0.3 \text{ V}$ as proper analysis range. Furthermore, we performed Impedance spectroscopy on further devices of a lower doping ratio (MR=0.004) and higher organic layer thickness (200 nm) as control experiment. The devices show rectification in I-V at all temperatures, and similar $d(1/C^2)/dV$ plots as previously are obtained, yielding thermal activation energies in the order of 24...25 meV (see below).

For the sake of clarity and readability, we did not show this data/plots in the Supporting Information of the manuscript, however, mentioned them in the caption of Fig. S3: "The uncertainty of the Arrhenius-type activation energy E_{act} determined by fitting Mott-Schottky plots is estimated to 5 meV by statistics on 3 ZnPc:F₆-TCNNQ diodes."

(4) On page 9 of the main text the authors argue that the higher activation energies

obtained for $C_{60}F_{36}$ is a reflection of a less favourable electrostatics. I found this somewhat surprising on the basis that for the fullerene dopant the electron wavefunction is less localised than for the F_6 -TCNNQ, i.e. one might have expected the Coulomb binding energy to be lower because of a larger distance between electron and hole? Why does this not manifest itself?

Reply: The reviewer raises a legitimate question. For sure, the spatial extent of the wavefunctions of dopant and host, i.e., the mutual distance of the electron and hole in the formed integer charge transfer complex (ICTC), determines the strength of the Coulomb binding in a first order. This is particularly expected for large molecules such as the fluorinated fullerene. However, as we pointed out in the discussion of the mentioned paragraph on pg. 9...10, the overall electrostatic environment is of equal importance for the final dissociation barrier for carrier release from the ICTC in a thin film. In that regard, the favorable impact of electrostatic interactions of charges with surrounded molecules' quadrupole and induced dipole moments on lowering the dissociation barrier for CT state separation in OPV D:A blends was previously shown (e.g. 10.1002/aenm.201601370, 10.1021/acsami.6b02851, 10.1021/jacs.5b02130), in particular, for flat molecules such as pentacene (10.1002/adfm.200901233, 10.1021/jp910005g).

For doped D:A systems, we believe that the same mechanisms control carrier release from ICTC, i.e., being determined by electrostatic moments on both surrounded dopant and host molecules. Similar to D/A interfaces, it is therefore reasonable to presume that the flat F_6 -TCNNQ dopant with its strong quadrupole moment yields favorable electrostatics regarding dissociation if compared to the more spherical $C_{60}F_{36}$ dopant. This circumstance we believe is manifested in the observed systematically lower E_{act} for F_6 -TCNNQ over $C_{60}F_{36}$.

REVIEWER #3:

Overall, to my point of view the authors provide a convincing model for mobile charge generation after the initial processes of ICTC formation. Their model is based on a solid footing of experimental data and the theoretical treatment thereof and will valuably contribute to a timely topic in the field of organic electronics. The data is well documented and the experimental information provided seems sufficient to me to reproduce the data. I would suggest publishing the present manuscript as it is. In their article "Elementary steps in electrical doping of organic semiconductors", M.L. Tietze et al. address the process of charge separation on molecularly doped organic semiconductors. While the process of charge transfer has been elucidated during the past years showing two competing processes of ground-state integer charge transfer (ICT) and host-dopant electronic wave-function hybridization, the subsequent step of carrier release, key for the generation of mobile electrons/holes upon n-/p-doping organic semiconductors, is still not fully understood today. In particular, it is not fully resolved how electron and hole after initial charge transfer can overcome their substantial Coulomb binding energy, i.e., how the thusly-formed integer charge transfer complexes (ICTCs) can thermally separate, as Coulomb binding contributes significantly more than kT can provide at room temperature.

The authors thoroughly address this important question both experimentally and by theoretical modelling. From temperature-dependent absorption spectroscopy on F_6 -TCNNQ-doped ZnPc and MeO-TPD films, two organic semiconductors with ionization energies lower than the electron affinity of the chosen dopant, they conclude that ICT does not significantly depend on the temperature down to 10K, i.e., doping freeze-out

is not due to less ionized acceptor molecules. From impedance spectroscopy (in a safe temperature range where deep traps are filled) they deduce Arrhenius-type activation energies of 20 meV and below, in particular, significantly lower than the expected Coulomb binding energy of an ICTC. The authors further stress that MeO-TPD, suggested to form amorphous films, has an even lower value than expectedly crystalline ZnPc, while the temperature dependence of the relative density of ionized acceptors is stronger for the latter. This fits well to the outcome of their modelling of ICTC occupation based on extensive (temperature-dependent) UPS data, where they conclude that energetic disorder is key for the dissociation of the ICTC, as it effectively lowers the activation energy. By their UPS analysis, the authors further provide evidence for temperature-dependent depletion widths, which are indicative for doping freeze-out.

Overall, to my point of view the authors provide a convincing model for mobile charge generation after the initial processes of ICTC formation. Their model is based on a solid footing of experimental data and the theoretical treatment thereof and will valuably contribute to a timely topic in the field of organic electronics. The data is well documented and the experimental information provided seems sufficient to me to reproduce the data. I would suggest publishing the present manuscript as it is.

Reply: We thank the reviewer for this evaluation. No further modifications are required.

REVIEWERS' COMMENTS:

Reviewer #1 (Remarks to the Author):

In the revised manuscript, the authors have addressed all the concerns in my original report. The work can now be accepted as is.

Reviewer #2 (Remarks to the Author):

The authors have provided a careful and reasonable response to my comments raised. However, they have only made minimal changes to my points 3 and 4 (as labelled in their response). I think it would be helpful to include these clarifications in the supplementary information as other readers might have similar questions.

REVIEWER #1:

In the revised manuscript, the authors have addressed all the concerns in my original report. The work can now be accepted as is.

Reply: We thank the reviewer for this evaluation. No further modifications are required.

REVIEWER #2:

The authors have provided a careful and reasonable response to my comments raised. However, they have only made minimal changes to my points 3 and 4 (as labelled in their response). I think it would be helpful to include these clarifications in the supplementary information as other readers might have similar questions.

Reply: As requested, we included these clarifications to the manuscript by following the reasoning of our previous response to Rev#2.

Regarding point (3), we added the following description to the Methods section (pg. 29) as well as extended Supplementary Figure 4:

“For Mott-Schottky analysis, measurement conditions which allow a determination of the active dopant density/activation were chosen for each material, i.e., the following two criteria are guaranteed: (1) Sufficient capacitive behavior of the device, i.e., ensuring measurement of the actual depletion capacitance C_d at the metal/semiconductor Schottky contact being neither superimposed by carrier injection when varying the voltage in $C(V)$ nor by the geometric capacitance of the organic film. Both aspects are achieved by choosing neither too high nor too low doping ratios. (2) Avoiding capacity contributions originating from charging and de-charging of trap states, which are characterized by different slopes in $C^{-2}(V)$ Mott-Schottky plots, i.e., causing kinks in those. Accordingly, bias voltages down to -1.0 V (p-MeO-TPD), -2.0 V (p-Ir(piq)₃), and $-0.2 \text{ V} < V < 0.3 \text{ V}$ (p-ZnPc) were used as analysis ranges. For the latter, Mott-Schottky analysis was performed on devices of varying doping ratio and organic layer thickness as control experiments (Supplementary Figure 3 and 4), from which an uncertainty of 5 meV for the Arrhenius-type activation energy E_{act} was estimated.”

Regarding point (4), we improved and sharpened the discussion and arguing on pg. 9/10 (yellow markings have been added):

“Additionally to (intrinsic) energetic disorder, the absolute dissociation probability is determined by electrostatic interactions in the specific D:A system. The ICTC binding energy E_{CT}^{b} is governed by the strong Coulomb attraction between electron and hole and further altered by interactions of the charges with quadrupole and induced dipole moments of surrounding molecules. [12] The contribution due to Coulomb attraction is the weaker the stronger the dopant and host wavefunctions localize and the more distant the associated hole and electron are. Thus, low activation energies are expected for large molecules such as fullerenes. However, doping C₆₀F₃₆ rather than F₆-TCNNQ into the investigated hosts yields systematically higher activation energies reaching even 54 meV for P5:C₆₀F₃₆ (cf. Table 1). This observation rather points to electrostatic interactions of the [D⁺A⁻] ICTC with surrounding molecules that actually favor the ICTC dissociation, an effect previously estimated to 0.1-0.5 eV. [12,48]

Electrostatic interactions were previously demonstrated to facilitate CT state dissociation in OPV D:A blends, [49] particularly for planar donor molecules with quadrupole moments such as P5.

[50,51] In the idealized case of flat D/A interfaces with C₆₀ as quadrupole-free acceptor, respective charge-quadrupole interactions were demonstrated to yield either repulsive or attractive forces, depending on the inter-molecular orientation of the donor, i.e., its quadrupole moment, with respect to the acceptor. [50,52] Similarly, it is hence reasonable to presume that the flat F₆-TCNNQ dopant with its strong quadrupole moment reduces ICTC binding energies compared to the more spherical C₆₀F₃₆ dopant; and indeed, E_{CT}^b (estimated from E_F pinning in UPS) is lower for ZnPc:F₆-TCNNQ (0.64 eV) by 0.24 eV than for P5:C₆₀F₃₆ (0.88 eV). [10] Since the intrinsic energetic disorder is similar for both hosts, we presume that the quadrupolar nature of the planar F₆-TCNNQ dopant is responsible for the systematically lower E_{act} compared to C₆₀F₃₆: 19 meV vs. 54 meV (P5:F₆-TCNNQ vs. P5:C₆₀F₃₆) and 9.1 meV vs. 16 meV (MeO-TPD:F₆-TCNNQ vs. MeO-TPD:C₆₀F₃₆). This direct comparison, thus, provides a crucial design rule for efficient host:dopant systems.”

Furthermore, we added (the now) Ref. 49 and 51 which were referred to in our previous response to Reviewer.

According to Nat. Comm. style guide-lines, we here provide the legend of Supplementary Movie 1:

Supplementary Movie 1 | Monte Carlo transport simulations on ZnPc:F₆-TCNNQ: Illustration of the simulated occupation probability W of sites with charge carriers (holes) in a 21x21x21 mesh with periodic boundary conditions and $MR=10^{-3}$ (10 dopants, without traps). Blue and orange dots represent the occupation probabilities of nearest neighbor (nn) and next nearest-neighbor (nn+1) sites of dopants, respectively, which positions are marked as red circles. For all other sites, the occupation probability is indicated by green dots. The occupation probability W increases with the plotted dot size each, here shown for a threshold of $W>10^{-4}$.

According to Nat. Comm. style guide-lines, we also updated Ref. 47:

[47] Pahner, P. Charge Carrier Trap Spectroscopy on Organic Hole Transport Materials. *Qucosa* (2017). <http://nbn-resolving.de/urn:nbn:de:bsz:14-qucosa-217882>